# Comparing the local information geometry of image representations

**David Lipshutz**[1,*]     **Jenelle Feather**[1,*]     **Sarah E. Harvey**[1]

**Alex H. Williams**[1,2]     **Eero P. Simoncelli**[1,2]

[1] Center for Computational Neuroscience, Flatiron Institute
[2] Center for Neural Science, New York University

{dlipshutz,jfeather,sharvey}@flatironinstitute.org,
{alex.h.williams,eero.simoncelli}@nyu.edu

## Abstract

We propose a framework for comparing a set of image representations (artificial or biological) in terms of their sensitivities to local distortions. We quantify the local geometry of a representation using the Fisher information matrix (FIM), a standard statistical tool for characterizing the sensitivity to local distortions of a stimulus, and use this as a substrate for a metric on the local geometry of representations in the vicinity of a base image. This metric may then be used to optimally differentiate a set of models, by optimizing for a pair of distortions that maximize the variance of the models under this metric. We use the framework to compare a set of simple models of the early visual system, identifying a novel set of image distortions that allow immediate comparison of the models by visual inspection. In a second example, we show that the method can reveal distinctions between standard and adversarially trained object recognition networks.

## 1  Introduction

Biological and artificial neural networks transform sensory stimuli into internal representations that support downstream tasks. Often, similarity between representations is quantified by measuring the alignment of their *global* geometric structure [1–5]. However, systems with similar global geometry can have strikingly different *local* geometries. The well-known occurrence of adversarial examples [6, 7] in object recognition systems provides an example: Even for systems that are broadly similar to each other and in agreement with human perception, carefully targeted experiments reveal small image distortions that are imperceptible to humans but result in reliable misclassification by a model.

How can we quantify and compare the local geometry of different image representations? A brute-force comparison clearly is prohibitive: the space of images is extremely high-dimensional, and the set of potential distortions equally high-dimensional. Estimating the local geometry of representations over a moderately dense sampling of this full set is impractical, and estimating human sensitivity to such a set is essentially impossible. As such, it is worthwhile to develop a method for judicious selection of stimulus distortions that can be used when comparing a set of models.

We take inspiration from Zhou et al. [8]. For a pair of models and a base image, they synthesize distortions along which the two models' sensitivities maximally disagree. This bears conceptual similarity to other methods that construct stimuli to optimally distinguish models [9, 10], and builds on earlier

---

[*]Equal contribution

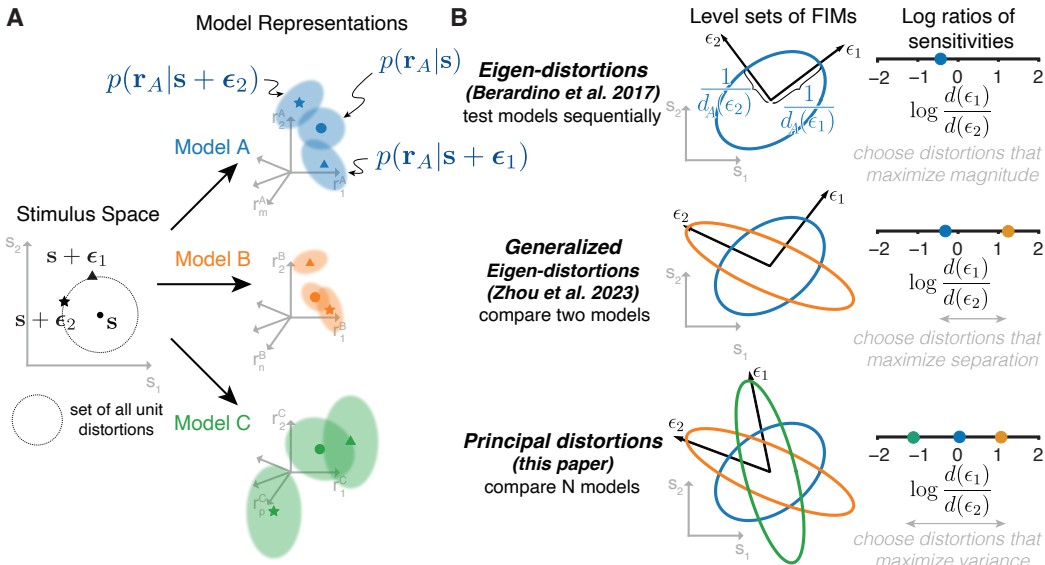

Figure 1: Comparing the local geometry of image representations. **A)** Each model maps a stimulus $\mathbf{s}$ to a distribution $p(\mathbf{r}|\mathbf{s})$ in representation space (biological neurons are noisy while deterministic models can be made stochastic by assuming additive Gaussian response noise). Classical signal detection theory posits that the sensitivity $d(\boldsymbol{\epsilon})$ of the representation to a distortion $\boldsymbol{\epsilon}$ depends on how much overlap there is between $p(\mathbf{r}|\mathbf{s})$ and $p(\mathbf{r}|\mathbf{s}+\boldsymbol{\epsilon})$, with less overlap indicating higher sensitivity [13]. **B)** Distortion sensitivity of each model may be mapped back to the stimulus domain via the FIM, which has been used previously to generate optimal stimuli for comparison to human perception. Specifically, the eigenvectors of a model FIM may be used to examine most/least model-sensitive stimuli [11], and the generalized eigenvectors of the ratio of model FIMs may be used to generate stimuli that best distinguish the sensitivities of two models [8]. Here, we show that these models have interpretations in terms of the log ratios of model sensitivities, and develop a generalized method that allows comparison of an arbitrary number of models, by selecting two stimulus distortions that maximize the variance of the log-ratio of sensitivities over models.

work that examined "eigen-distortions" along which individual models are maximally/minimally sensitive [11]. Specifically, they measure the local sensitivity of a model in terms of its Fisher Information matrix (FIM) [12], a classical tool from statistical estimation theory, and choose the pair of "generalized eigen-distortions" that maximize/minimize the ratio of the two models' sensitivities. Once these image distortion have been computed, they may be added in varying amounts to a base image to determine the level at which they become visible to a human. These measured human sensitivities can then be compared to those of the models, with the goal of identifying which model is better aligned with the local geometry of the human visual system. However, when comparing more than two models, there is no principled method for selecting the distortions.

Here we define a novel metric for comparing model representations in terms of their relative sensitivities to a pair of local image distortions. We then use this metric to generate a pair of distortions that maximize the variance of two or more models. In analogy with principal component analysis, we refer to these as "principal distortions". We apply our method to a nested set of hand-crafted models of the early visual system, and to a set of standard and adversarially-trained visual neural network models. In both cases, we illustrate how the method generates novel distortions that highlight differences between models. For additional details and results, see the full paper version of this extended abstract [14].

## 2 Problem statement and methods

Given a collection of stochastic image representations, our goal is to develop a method for comparing the local geometry in the vicinity of image $\mathbf{s}$ across these image representations. We assume that each representation has an associated conditional density $p(\mathbf{r}|\mathbf{s})$, where $\mathbf{s} \in \mathbb{R}^K$ is a vector of image pixels and $\mathbf{r}$ is a stochastic response (e.g., neuronal firing rates or deterministic model responses with additive response noise, see Fig. 1A). Associated with the conditional density is the Fisher-Rao

metric [15, 16] (Fig. 1B), a Riemannian metric on the stimulus space defined in terms of the Fisher information matrix (FIM) [12]

$$\boldsymbol{I}(\mathbf{s}) := \mathbb{E}_{\mathbf{r} \sim p(\mathbf{r}|\mathbf{s})} \left[ \nabla_{\mathbf{s}} \log p(\mathbf{r}|\mathbf{s}) \nabla_{\mathbf{s}} \log p(\mathbf{r}|\mathbf{s})^{\top} \right].$$

The *sensitivity* of the representation to distortions of stimulus $\mathbf{s}$ in the direction $\boldsymbol{\epsilon}$ can be expressed as:

$$d(\boldsymbol{\epsilon}) = d(\mathbf{s}; \boldsymbol{\epsilon}) := \sqrt{\boldsymbol{\epsilon}^{\top} \boldsymbol{I}(\mathbf{s}) \boldsymbol{\epsilon}}. \tag{1}$$

A comprehensive comparison of the local geometries of two or more image representations is impractical. Therefore, it is useful to develop a method for optimally choosing image distortions along which to compare image representations. Berardino et al. [11] proposed computing the extremal eigenvectors of model FIMs (termed "eigen-distortions", Fig. 1B) and comparing them to human visual sensitivities. However, if the eigen-distortions of two models are similar, they will not be useful in distinguishing the models, since they will be insensistive to differences in the non-extremal eigenvectors.

**Comparing two image representations**   Zhou et al. [8] proposed comparing two image representations $A$ and $B$ along distortions in which their local sensitivities maximally differ. Specifically, they chose distortions to extremize the generalized Rayleigh quotient:

$$\boldsymbol{\epsilon}_1 = \arg\max_{\boldsymbol{\epsilon}} \frac{\boldsymbol{\epsilon}^{\top} \boldsymbol{I}_A(\mathbf{s}) \boldsymbol{\epsilon}}{\boldsymbol{\epsilon}^{\top} \boldsymbol{I}_B(\mathbf{s}) \boldsymbol{\epsilon}}, \qquad\qquad \boldsymbol{\epsilon}_2 = \arg\min_{\boldsymbol{\epsilon}} \frac{\boldsymbol{\epsilon}^{\top} \boldsymbol{I}_A(\mathbf{s}) \boldsymbol{\epsilon}}{\boldsymbol{\epsilon}^{\top} \boldsymbol{I}_B(\mathbf{s}) \boldsymbol{\epsilon}}.$$

Since these distortions correspond to the extremal eigenvectors of the generalized eigenvalue problem $\boldsymbol{I}_A(\mathbf{s})\boldsymbol{\epsilon} = \lambda \boldsymbol{I}_B(\mathbf{s})\boldsymbol{\epsilon}$, we refer to them as "generalized eigen-distortions" (Fig. 1B). However, this method is limited to comparisons of pairs of models, or a single model to the average of other models.

**Comparing many image representations**   The generalized eigenvalue problem induces a metric between image representations, which can be used to optimally choose image distortions for distinguishing more than two models. Specifically, up to permutation, we can express the pair of generalized eigen-distortions as the solution to the following optimization problem (Appx. A):

$$\{\boldsymbol{\epsilon}_1, \boldsymbol{\epsilon}_2\} = \arg\max_{\boldsymbol{\epsilon}, \boldsymbol{\epsilon}'} m_{\boldsymbol{\epsilon}, \boldsymbol{\epsilon}'}(A, B), \quad \text{for} \quad m_{\boldsymbol{\epsilon}, \boldsymbol{\epsilon}'}(A, B) := \left| \log \frac{d_A(\boldsymbol{\epsilon})}{d_A(\boldsymbol{\epsilon}')} - \log \frac{d_B(\boldsymbol{\epsilon})}{d_B(\boldsymbol{\epsilon}')} \right|. \tag{2}$$

For any pair of distortions $\boldsymbol{\epsilon}, \boldsymbol{\epsilon}'$, the function $m_{\boldsymbol{\epsilon}, \boldsymbol{\epsilon}'}(\cdot, \cdot)$ defines a *metric* on the local geometry of image representations: it is non-negative, symmetric, and obeys the triangle inequality. The extremal distortions have several appealing properties: (i) they are invariant to arbitrary rescaling of the FIM ($\boldsymbol{I}(\mathbf{s}) \mapsto c\boldsymbol{I}(\mathbf{s})$ for any $c > 0$), of either model; (ii) they are invariant to permutations ($\boldsymbol{\epsilon} \leftrightarrow \boldsymbol{\epsilon}'$); (iii) when $\boldsymbol{\epsilon}, \boldsymbol{\epsilon}'$ are the generalized eigen-distortions, the metric is an approximation of the Fisher-Rao distance between mean-zero Gaussian distributions with covariances $\boldsymbol{I}_A(\mathbf{s})$ and $\boldsymbol{I}_B(\mathbf{s})$ (Appx. B); and (iv) the metric compares stochastic representations back in stimulus space, which avoids the problem of having to align two representational spaces via a nuisance transformation [17].

We can generalize the result to optimize a pair of image distortions for distinguishing $N > 2$ image representations $A_1, \ldots, A_N$. In particular, we choose $\boldsymbol{\epsilon}_1, \boldsymbol{\epsilon}_2$ to maximize the sum of the squares of all pairwise differences between the log sensitivity ratios, which is equivalent to maximizing the *variance* of the image representations' log sensitivity ratios:

$$\{\boldsymbol{\epsilon}_1, \boldsymbol{\epsilon}_2\} = \arg\max_{\boldsymbol{\epsilon}, \boldsymbol{\epsilon}'} \sum_{n=1}^{N} \left| \log \frac{d_{A_n}(\boldsymbol{\epsilon})}{d_{A_n}(\boldsymbol{\epsilon}')} - \frac{1}{N} \sum_{m=1}^{N} \log \frac{d_{A_m}(\boldsymbol{\epsilon})}{d_{A_m}(\boldsymbol{\epsilon}')} \right|^2$$

We refer to $\{\boldsymbol{\epsilon}_1, \boldsymbol{\epsilon}_2\}$ as the "principal distortions" of the models, analogous to principal component analysis (Fig. 1B). For a gradient-based optimization algorithm, see Appx. C.

## 3   Experiments

As a demonstration of our method, we generated principal distortions for computational models previously proposed to capture aspects of the human visual system. All models were implemented in PyTorch and simulations were performed on a NVIDIA RTX A6000 GPU. As the models are deterministic, we follow the assumptions of [11] and calculate the FIM by assuming the network output is corrupted by additive Gaussian noise, in which case $\boldsymbol{I}(\mathbf{s}) = \boldsymbol{J}_f(\mathbf{s})^{\top} \boldsymbol{J}_f(\mathbf{s})$, where $\boldsymbol{J}_f(\mathbf{s})$ is the Jacobian of the model $\boldsymbol{f}(\cdot)$ at $\mathbf{s}$.

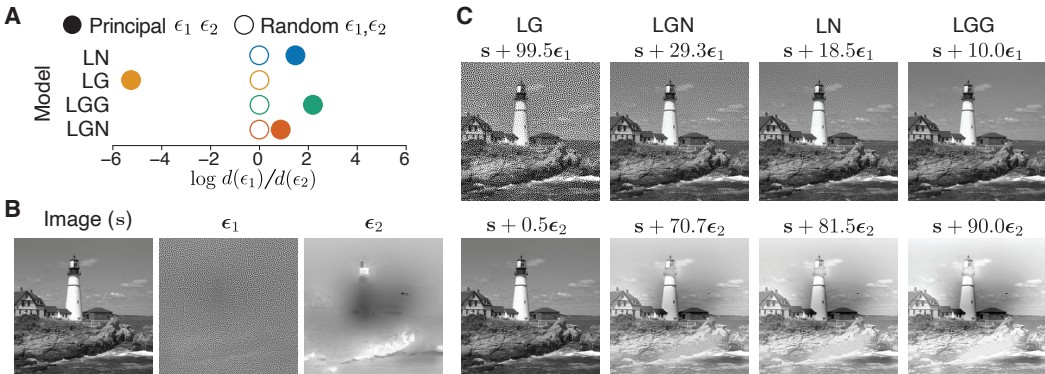

Figure 2: Principal distortions of a set of early visual models. **A)** Log sensitivity ratios of principal distortions and two random distortions for each of the four models. Principal distortions (filled circles) separate the log ratios, while random distortions (hollow circles) do not. **B)** Natural image **s** and corresponding optimized principal distortions $\{\epsilon_1, \epsilon_2\}$. **C)** Natural image corrupted by principal distortions, with each pair scaled so as to be equally detectable by the corresponding model. Models are ordered by the log ratio of their sensitivities (panel A). If a model's thresholds are comparable to human thresholds, the two scaled distortions should be equally visible in the top and bottom images. Note: Distorted images are best viewed at high resolution.

**Early visual models** We generated principal distortions for a nested family of models designed to capture the early visual structure and computations (Fig. 2A,B). The full model (LGN) contains two parallel cascades representing ON and OFF channels, rectification, and both luminance and contrast gain control nonlinearities. The other models are reduced versions of this model. LGG removes the OFF channel, LG additionally removes the contrast gain control, and LN removes both gain controls. The filter size, amplitude, and normalization values were previously fit separately for each model to predict human distortion ratings [11].

To provide a qualitative comparison of each model's sensitivities to human distortion sensitivity, we adjusted the relative scaling of the principal distortion so as to be equally detectable by that model, while constraining the sum of the norms of the two distortions to be 100 (Fig. 2C). If a model's thresholds are comparable to human thresholds, then these rescaled distortions should be equally detectable when added to the image **s**. Visual inspection of these images reveals that both distortions are visible when rescaled for the LGN model and the LN model, suggesting that these models are closest to human distortion thresholds. For LG, the scaled $\epsilon_2$ distortion is not visible, while the scaled $\epsilon_1$ distortion is immediately apparent, suggesting a strong mismatch with human observer thresholds. The same is true of the LGG model, with the roles of the two distortions swapped. These qualitative observations are consistent with the results of [11], in which experiments on eigen-distortions suggested that the LGN model was the best of these models in terms of consistency with human distortion sensitivity. Formal perceptual experiments could be performed in the future to explicitly quantify the visibility of the principal distortions arising from our analysis. Crucially, measurements of human perceptual sensitivity are costly and our proposed method only requires measuring sensitivity to two distortions (in contrast to eight eigen-distortions).

**Deep neural networks** Deep Neural Networks (DNNs), originally developed for object recognition, have also been examined as models of the primate visual system [2, 18, 19]. A plethora of models, varying in architecture and training techniques, have been proposed, but many of these models perform quite similarly on behavioral tasks or neural benchmarks [18, 20, 21]. This situation offers a well-aligned opportunity for our principal distortion method. As a demonstration of the method, we measured the FIM of a set of layers from two different architectures (AlexNet [22] and ResNet50 [23]) and two different training procedures (standard vs. adversarial training, AT) with $\ell_2$-norm perturbations [24, 25], for a total of $N = 28$ different model representations. AT networks were initially developed for engineering purposes to reduce the vulnerability of the models to adversarial examples [26, 27], but previous work has also found that representations in AT networks are more aligned with those of biological systems [26, 25, 28]. As adversarial examples are constrained to be very small perturbations, it seems plausible that the local geometry for AT models would differ from their standard counterparts.

We show an example principal distortion generated for the set of layers from the four DNNs (Fig. 3). The hierarchical structure of the models is reflected in the log ratios of the sensitivities, where early layers of the models (smaller dots) are closer together in the metric space, and later layers of the models (larger dots) are pushed further away. Overall, most layers of the standard networks are more sensitive to the distortion that appears as unstructured noise, while most layers of the AT networks are more sensitive to the structured distortion. This qualitative example provides a demonstration that our method can be used to separate collections of similar models, and points to its utility in probing complex high-level representations.

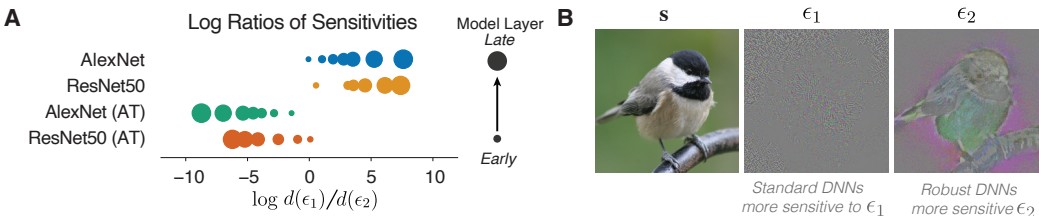

Figure 3: Principal distortions for standard and adversarially trained (AT) DNNs. **A)** Log sensitivity ratios of principal distortions at a base image. **B)** Base image and the generated principal distortions. Distortion $\epsilon_1$ appears as less structured noise, and both AlexNet and ResNet50 standard networks are more sensitive to this perturbation, while the AT DNNs are more sensitive to $\epsilon_2$ which focuses color changes around the content of the image, suggesting that the differences in local sensitivities of these networks depend more on differences in training procedure than architecture.

## 4  Discussion

We introduced a metric for image representations that captures the local geometry, and used it to synthesize "principal distortions" that maximize the variance of this metric over a set of models. When applied to hand-engineered models of the early visual system and to standard and AT DNNs, our approach produced novel distortions for distinguishing the corresponding models.

The relation to the Fisher-Rao metric (Appx. B) suggests an extension for synthesizing more than two distortions. Additionally, there is a natural extension to continuous families of models. We plan to explore these directions in future work.

These distortions provide an efficient method for comparing computational models with human observers, for whom the experimental time for acquiring responses to stimuli is generally severely limited. The optimized distortions are a parsimonious choice of stimuli, that can be readily incorporated into psychophysics experiments, whose results can guide further model development.

## Acknowledgments and Disclosure of Funding

The Flatiron Institute is a division of the Simons Foundation. The computations reported in this paper were performed using resources made available by the Flatiron Institute. We additionally thank David Brainard, Thomas Yerxa, and the Laboratory for Computational Vision for their feedback.

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

## A    Deriving the metric from a generalized eigenvalue problem

Our starting point is the following expression for the generalized eigen-distortions:

$$\boldsymbol{\epsilon}_1 = \arg\max_{\boldsymbol{\epsilon}} \frac{\boldsymbol{\epsilon}^\top \boldsymbol{I}_A(\mathbf{s})\boldsymbol{\epsilon}}{\boldsymbol{\epsilon}^\top \boldsymbol{I}_B(\mathbf{s})\boldsymbol{\epsilon}}, \qquad\qquad \boldsymbol{\epsilon}_2 = \arg\min_{\boldsymbol{\epsilon}} \frac{\boldsymbol{\epsilon}^\top \boldsymbol{I}_A(\mathbf{s})\boldsymbol{\epsilon}}{\boldsymbol{\epsilon}^\top \boldsymbol{I}_B(\mathbf{s})\boldsymbol{\epsilon}}.$$

From the definition for the sensitivity $d(\boldsymbol{\epsilon})$ in Equation 1 and the monotonicty of the square root and logarithm operations, we can express the generalized eigen-distortions as follows:

$$\boldsymbol{\epsilon}_1 = \arg\max_{\boldsymbol{\epsilon}} \log \frac{d_A(\boldsymbol{\epsilon})}{d_B(\boldsymbol{\epsilon})}, \qquad\qquad \boldsymbol{\epsilon}_2 = \arg\min_{\boldsymbol{\epsilon}} \log \frac{d_A(\boldsymbol{\epsilon})}{d_B(\boldsymbol{\epsilon})}.$$

Up to permutation, these distortions can be expressed as the solution of the following optimization problem:

$$\{\boldsymbol{\epsilon}_1, \boldsymbol{\epsilon}_2\} = \arg\max_{\boldsymbol{\epsilon},\boldsymbol{\epsilon}'} \left| \log \frac{d_A(\boldsymbol{\epsilon})}{d_B(\boldsymbol{\epsilon})} - \log \frac{d_A(\boldsymbol{\epsilon}')}{d_B(\boldsymbol{\epsilon}')} \right|,$$

which can be rearranged to produce Equation 2.

## B    Relation to the Fisher-Rao metric

The Fisher-Rao distance between two mean zero Gaussian with covariances $\boldsymbol{A}$ and $\boldsymbol{B}$ is equal to

$$\delta^2(\boldsymbol{A}, \boldsymbol{B}) := \|\log(\boldsymbol{B}^{-1/2}\boldsymbol{A}\boldsymbol{B}^{-1/2})\|_F^2 = \sum_{i=1}^{K}(\log \lambda_i)^2,$$

where $\{\lambda_i\}$ denote the eigenvalues of the generalized eigenvalue problem $\boldsymbol{A}\boldsymbol{v} = \lambda\boldsymbol{B}\boldsymbol{v}$. We'd like a metric that's invariant to scaling $\boldsymbol{A}$ or $\boldsymbol{B}$, which suggests using the metric

$$\gamma^2(\boldsymbol{A}, \boldsymbol{B}) = \min_{c_A, c_B > 0} \delta^2(c_A \boldsymbol{A}, c_B \boldsymbol{B}) = \min_{c \in \mathbb{R}} \sum_{i=1}^{K}(c + \log \lambda_i)^2 = K \mathrm{Var}(\{\log \lambda_i\}),$$

where the final equality uses the fact that the optimal $c$ is the mean of $\{-\log \lambda_i\}$.

Using the facts that

$$K\text{Var}(\{\log \lambda_i\}) = \frac{1}{2K}\sum_{i=1}^{K}\sum_{j=1}^{K}(\log \lambda_i - \log \lambda_j)^2,$$

and $(\log \lambda_i - \log \lambda_j)^2 \le (\log \lambda_1 - \log \lambda_K)^2$ for all $i, j$, we have

$$\frac{1}{K}(\log \lambda_1 - \log \lambda_K)^2 \le \gamma^2(\boldsymbol{A}, \boldsymbol{B}) \le \frac{K-1}{2}(\log \lambda_1 - \log \lambda_K)^2.$$

When $\boldsymbol{A} = \boldsymbol{I}_A$ and $\boldsymbol{B} = \boldsymbol{I}_B$, then $d_A(\boldsymbol{\epsilon}) = \sqrt{\boldsymbol{\epsilon}^\top \boldsymbol{A}\boldsymbol{\epsilon}}$ and $d_B(\boldsymbol{\epsilon}) = \sqrt{\boldsymbol{\epsilon}^\top \boldsymbol{B}\boldsymbol{\epsilon}}$. If $\boldsymbol{\epsilon}_1$ and $\boldsymbol{\epsilon}_K$ denote the extremal generalized eigenvectors associated with $\lambda_1$ and $\lambda_K$, respectively, then

$$\log \lambda_1 = 2\log \frac{d_A(\boldsymbol{\epsilon}_1)}{d_B(\boldsymbol{\epsilon}_1)}, \qquad\qquad \log \lambda_K = 2\log \frac{d_A(\boldsymbol{\epsilon}_K)}{d_B(\boldsymbol{\epsilon}_K)}.$$

Therefore,

$$\frac{2}{\sqrt{K}}\left|\log \frac{d_A(\boldsymbol{\epsilon}_1)}{d_B(\boldsymbol{\epsilon}_1)} - \log \frac{d_A(\boldsymbol{\epsilon}_K)}{d_B(\boldsymbol{\epsilon}_K)}\right| \le \gamma(\boldsymbol{A}, \boldsymbol{B}) \le \sqrt{2(K-1)}\left|\log \frac{d_A(\boldsymbol{\epsilon}_1)}{d_B(\boldsymbol{\epsilon}_1)} - \log \frac{d_A(\boldsymbol{\epsilon}_K)}{d_B(\boldsymbol{\epsilon}_K)}\right|,$$

and so

$$\frac{2}{\sqrt{K}}m_{\boldsymbol{\epsilon}_1,\boldsymbol{\epsilon}_K}(\boldsymbol{I}_A, \boldsymbol{I}_B) \le \gamma(\boldsymbol{A}, \boldsymbol{B}) \le \sqrt{2(K-1)}m_{\boldsymbol{\epsilon}_1,\boldsymbol{\epsilon}_K}(\boldsymbol{I}_A, \boldsymbol{I}_B).$$

## C   Computing the top two optimal distortions

Suppose we have $N$ models with sensitivities $\{d_n(\boldsymbol{\epsilon})\}$. The optimal distortions $\{\boldsymbol{\epsilon}_1, \boldsymbol{\epsilon}_2\}$ are solutions to the optimization problem

$$\arg\max_{\boldsymbol{\epsilon}_1, \boldsymbol{\epsilon}_2} L(\boldsymbol{\epsilon}_1, \boldsymbol{\epsilon}_2), \qquad L(\boldsymbol{\epsilon}_1, \boldsymbol{\epsilon}_2) := \sum_{n=1}^{N}\left\{\log \frac{d_n(\boldsymbol{\epsilon}_1)}{d_n(\boldsymbol{\epsilon}_2)} - \frac{1}{N}\sum_{m=1}^{N}\log \frac{d_m(\boldsymbol{\epsilon}_1)}{d_m(\boldsymbol{\epsilon}_2)}\right\}^2.$$

Differentiating $L$ with respect to $\boldsymbol{\epsilon}_1$ yields

$$\nabla_{\boldsymbol{\epsilon}_1} L(\boldsymbol{\epsilon}_1, \boldsymbol{\epsilon}_2) = 2\sum_{n=1}^{N}\left\{\log \frac{d_n(\boldsymbol{\epsilon}_1)}{d_n(\boldsymbol{\epsilon}_2)} - \frac{1}{N}\sum_{m=1}^{N}\log \frac{d_m(\boldsymbol{\epsilon}_1)}{d_m(\boldsymbol{\epsilon}_2)}\right\}\left\{\frac{\boldsymbol{I}_n(\mathbf{s})\boldsymbol{\epsilon}_1}{d_n^2(\boldsymbol{\epsilon}_1)} - \frac{1}{N}\sum_{m=1}^{N}\frac{\boldsymbol{I}_m(\mathbf{s})\boldsymbol{\epsilon}_1}{d_m^2(\boldsymbol{\epsilon}_1)}\right\}$$

$$= \sum_{n=1}^{N}\left\{\log \frac{d_n^2(\boldsymbol{\epsilon}_1)}{d_n^2(\boldsymbol{\epsilon}_2)} - \frac{1}{N}\sum_{m=1}^{N}\log \frac{d_m^2(\boldsymbol{\epsilon}_1)}{d_m^2(\boldsymbol{\epsilon}_2)}\right\}\left\{\frac{\boldsymbol{I}_n(\mathbf{s})\boldsymbol{\epsilon}_1}{d_n^2(\boldsymbol{\epsilon}_1)} - \frac{1}{N}\sum_{m=1}^{N}\frac{\boldsymbol{I}_m(\mathbf{s})\boldsymbol{\epsilon}_1}{d_m^2(\boldsymbol{\epsilon}_1)}\right\},$$

where we have used the fact that

$$\nabla_{\boldsymbol{\epsilon}}\log d(\boldsymbol{\epsilon}) = \frac{1}{2}\nabla_{\boldsymbol{\epsilon}}\log(\boldsymbol{\epsilon}^\top \boldsymbol{I}\boldsymbol{\epsilon}) = \frac{\boldsymbol{I}\boldsymbol{\epsilon}}{\boldsymbol{\epsilon}^\top \boldsymbol{I}\boldsymbol{\epsilon}} = \frac{\boldsymbol{I}\boldsymbol{\epsilon}}{d^2(\boldsymbol{\epsilon})}.$$

Similarly, differentiating $L$ with respect to $\boldsymbol{\epsilon}_2$ yields:

$$\nabla_{\boldsymbol{\epsilon}_2} L(\boldsymbol{\epsilon}_1, \boldsymbol{\epsilon}_2) = -\sum_{n=1}^{N}\left\{\log \frac{d_n^2(\boldsymbol{\epsilon}_1)}{d_n^2(\boldsymbol{\epsilon}_2)} - \frac{1}{N}\sum_{m=1}^{N}\log \frac{d_m^2(\boldsymbol{\epsilon}_1)}{d_m^2(\boldsymbol{\epsilon}_2)}\right\}\left\{\frac{\boldsymbol{I}_n(\mathbf{s})\boldsymbol{\epsilon}_2}{d_n^2(\boldsymbol{\epsilon}_2)} - \frac{1}{N}\sum_{m=1}^{N}\frac{\boldsymbol{I}_m(\mathbf{s})\boldsymbol{\epsilon}_2}{d_m^2(\boldsymbol{\epsilon}_2)}\right\}.$$

Combining, we have the following gradient-based optimization algorithm.

**Algorithm 1:** Computing the top two distortions

1: **Input:** Positive definite $D \times D$ matrices $\boldsymbol{I}_1, \ldots, \boldsymbol{I}_N$, learning rate $\eta > 0$
2: **Initialize:** $\boldsymbol{\epsilon}_1, \boldsymbol{\epsilon}_2 \in \mathbb{R}^D$
3: **while** not converged **do**
4:    **for** $n = 1, \ldots, N$ **do**
5:       $\boldsymbol{v}_1(n) \leftarrow \boldsymbol{I}_n \boldsymbol{\epsilon}_1$
6:       $\boldsymbol{v}_2(n) \leftarrow \boldsymbol{I}_n \boldsymbol{\epsilon}_2$
7:       $d_1^2(n) \leftarrow \langle \boldsymbol{\epsilon}_1, \boldsymbol{v}_1(n) \rangle$
8:       $d_2^2(n) \leftarrow \langle \boldsymbol{\epsilon}_2, \boldsymbol{v}_2(n) \rangle$
9:       $\boldsymbol{u}_1(n) = \boldsymbol{v}_1(n)/d_1^2(n)$
10:      $\boldsymbol{u}_2(n) = \boldsymbol{v}_2(n)/d_2^2(n)$
11:      $r(n) \leftarrow \log d_1^2(n) - \log d_2^2(n)$
12:    **end for**
13:    $\bar{\boldsymbol{u}}_1 \leftarrow \mathrm{mean}(\boldsymbol{u}_1(n))$
14:    $\bar{\boldsymbol{u}}_2 \leftarrow \mathrm{mean}(\boldsymbol{u}_2(n))$
15:    $\bar{r} \leftarrow \mathrm{mean}(r(n))$
16:    $\Delta \boldsymbol{\epsilon}_1 \leftarrow \sum_{n=1}^N [r(n) - \bar{r}] [\boldsymbol{u}_1(n) - \bar{\boldsymbol{u}}_1]$
17:    $\Delta \boldsymbol{\epsilon}_2 \leftarrow - \sum_{n=1}^N [r(n) - \bar{r}] [\boldsymbol{u}_2(n) - \bar{\boldsymbol{u}}_2]$
18:    $\boldsymbol{\epsilon}_1 \leftarrow \boldsymbol{\epsilon}_1 + \eta \Delta \boldsymbol{\epsilon}_1$
19:    $\boldsymbol{\epsilon}_2 \leftarrow \boldsymbol{\epsilon}_2 + \eta \Delta \boldsymbol{\epsilon}_2$
20:    $\boldsymbol{\epsilon}_1 \leftarrow \boldsymbol{\epsilon}_1 / \|\boldsymbol{\epsilon}_1\|$
21:    $\boldsymbol{\epsilon}_2 \leftarrow \boldsymbol{\epsilon}_2 / \|\boldsymbol{\epsilon}_2\|$
22: **end while**

