# OpenReview forum: "Comparing the local information geometry of image representations"
_NeurIPS.cc/2024/Workshop/UniReps — UniReps_

### Official Review · Reviewer_tbqt · 2024-10-04
**Local variations based image representation comparison**

**Rating:** 6
**Confidence:** 3

**Review:**

The paper introduces a novel metric for comparing image representations across different models, leveraging the Fisher information matrix to assess sensitivity to local distortions in images. The theoretical foundation appears robust, and the introduction effectively situates the work within the context of related studies.

However, there are several areas that could be improved. The figures in the experimental section are not referenced in the main text, which affects the clarity and flow of the paper. Additionally, the experiments themselves are fairly simple and lack depth. Pushing the evaluation toward more challenging and varied scenarios, such as comparing early visual models with deep neural networks, would provide a stronger validation of the proposed metric. The experimental results also lack deeper insights, leaving the reader wanting more exploration of the metric's implications and applications.

Despite these issues, I believe the core idea has significant potential, and with further development and more rigorous experimentation, it could lead to interesting applications in image representation analysis.

---

### Official Review · Reviewer_XiUu · 2024-10-05
**A decent idea on improving the comparison of a set of image representations using local distortions**

**Rating:** 7
**Confidence:** 3

**Review:**

The paper provides a brief description of a framework for comparing a set of image representations using local distortions. The paper builds on the generalized eigen-distortions using a novel metric to generate what they called "“principal distortions".

Pros:
- The paper clearly explains the details of the framework.
- The framework is tested both on simple models of the early visual system and deep neural networks.

Cons:
- I understand that this is an abstract paper, however the paper does not provide enough results to confirm that the idea is actually an improvement over previous methods. Only a couple of visuals are provided.

---

### Official Review · Reviewer_ccvF · 2024-10-05
**Review of Submission51**

**Rating:** 6
**Confidence:** 3

**Review:**

This paper introduces a framework to compare image representations by analyzing their sensitivities to local distortions based on the Fisher Information Matrix. It defines "principal distortions" to highlight differences between multiple models and applies the method to both visual system models and deep neural networks, showing its utility in distinguishing models' behaviors.

Strengths

- The method and problems are clearly and rigorously stated.
- The extension of eigen-distortions to the multi-model case is valuable and well motivated.

Weaknesses:

- It is not always clear what insights should be drawn from the experimental section: in particular, results on early visual models are not well commented and the key takeaway does not emerge clearly.
- In the experiment on deep networks, it is not clear how the authors quantify the structuredness of noise, and why adversarially trained networks should be expected to differ from standard ones in this regard.

---

### Decision · Program_Chairs · 2024-10-10

**Decision:**

Accept

**Comment:**

In light of the positive reviewers' feedback and relevancy of the submission, we are pleased to accept this paper for presentation at UniReps 2024. We kindly ask the authors to incorporate the reviewers' suggestions and feedback in the final camera-ready version of the manuscript.